# Development of a curricular thread to foster medical students' critical reflection and promote action on climate change, health, and equity

Trisha Dalapati[1], Emily J. Alway[1], Sneha Mantri[2], Phillip Mitchell[3], Ian A. George[1], Samantha Kaplan[4], Kathryn M. Andolsek[5], J. Matthew Velkey[6], Jennifer Lawson[7], Andrew J. Muzyk[8]*

1 Duke University School of Medicine, Durham, NC, United States of America, 2 Program in Medical Humanities, Trent Center for Bioethics, Humanities & History of Medicine, Duke University School of Medicine, Durham, NC, United States of America, 3 Interprofessional Education Program, Western University of Health Sciences, Lebanon, OR, United States of America, 4 Duke University Medical Center Library & Archives, Duke University School of Medicine, Durham, NC, United States of America, 5 Department of Family Medicine and Community Health, Duke University School of Medicine, Durham, NC, United States of America, 6 Practice of Medical Education, Department of Cell Biology, Duke University School of Medicine, Durham, NC, United States of America, 7 Trent Center for Bioethics, Humanities & History of Medicine, Duke University School of Medicine, Durham, NC, United States of America, 8 Practice of Medical Education, Duke University School of Medicine, Durham, NC, United States of America

* Andrew.Muzyk@duke.edu

## Abstract

### Introduction

Due to the health consequences arising from climate change, medical students will inevitably interact with affected patients during their training and careers. Accordingly, medical schools must incorporate education on the impacts of climate change on health and equity into their curricula. We created a curricular thread called "Climate Change, Health, and Equity" in the first-year preclinical medical program to teach foundational concepts and foster self-reflection and critical consciousness.

### Methods

The authors developed a continuum of practice including administrators, educators and faculty members, students, and community partners to plan and design curricular activities. First-year medical students at Duke University School of Medicine participated in seven mandatory foundational lectures and two experiential learning opportunities in the local community. Following completion of activities, students wrote a critical reflection essay and completed a self-directed learning exercise. Essays were evaluated using the REFLECT rubric to assess if students achieved critical reflection and for thematic analysis by Bloom's Taxonomy.

**Funding:** The author(s) received no specific funding for this work.

**Competing interests:** No authors have competing interests

## Results

All students (118) submitted essays. A random sample of 30 (25%) essays underwent analysis. Evaluation by the REFLECT rubric underscored that all students were reflecting or critically reflecting on thread content. Thematic analysis highlighted that all students (30/30, 100%) were adept at identifying new areas of medical knowledge and connecting concepts to individual experiences, institutional practices, and public health and policy. Most students (27/30; 90%) used emotionally laden words, expressing negative feelings like frustration and fear but also positive sentiments of solidarity and hope regarding climate change and effects on health. Many students (24/30; 80%) expressed actionable items at every level including continuing self-directed learning and conversing with patients, minimizing healthcare waste, and advocating for climate-friendly policies.

## Conclusion

After participating in the curricular thread, most medical students reflected on cognitive, affective, and actionable aspects relating to climate change, health, and equity.

## Introduction

Beyond learning about the conditions of the human body, diagnostics, and treatment modalities, medical students are well-positioned to examine how patients' health is affected by the surrounding environment, including the climate, geography, and healthcare inequities in the community. With the inevitable health consequences arising from climate change, medical students will interact with affected patients during their training and careers. Students are increasingly aware that the very healthcare systems they are members of contribute significantly to greenhouse gas emissions underlying anthropogenic climate change [1, 2]. Moreover, while all individuals may be affected, communities that are historically marginalized, medically underserved, and older are at higher risk of experiencing the health impacts of climate change due to structural inequities [3, 4]. At present, medical schools have both an opportunity and a responsibility to promote action-oriented learning in undergraduate medical education (UME) at the intersection of climate change and health.

In 2015, over one hundred health professional schools signed the Health Educator's Climate Commitment, an international pledge to train the next generation of health professionals to care for patients impacted by climate change [5]. Medical schools are responding by incorporating climate change topics into UME. From 2019 to 2022, the Liaison Committee on Medical Education reported that the percentage of medical schools providing climate change instruction more than doubled from 27% to 55% [6]. In addition to learning the facts and diagnoses related to climate change, medical students are uniquely situated to be "agents of change" by identifying, researching, and advocating for climate resilience, environmental justice, and health equity [7, 8]. Education on climate change's effects on health has the potential to be transformative, allowing students and educators to synthesize factual information, reflect on content beyond the classroom, and evoke commitment to personal and professional behavior change.

However, UME faces challenges with incorporating climate concepts into curricula. Recurring barriers include competition for time, space, and faculty in the existing curriculum [9, 10]. Climate change and its health impacts are also new topics for medical educators. Building

teaching capacity, assessing resources, and integrating new material into existing content necessitate time and associated administrative and financial support [9–12]. Finally, there is a dearth of robust assessments to evaluate student learning. Assessments are critical as they signal learning priorities to students, confirm knowledge acquisition and behavior change, and facilitate sharing of successful interventions across programs and institutions [10, 11, 13, 14]. Thought leaders have recommended learning objectives for climate change and health education, and suggested when and how to deliver content and assessments within existing educational frameworks [13–18].

In fall 2022, faculty members and trainees met to address the lack of UME at our institution, Duke University School of Medicine (DUSOM), focused on climate change and its health impacts. Subsequently, we created a community of practice (CoP) of educators and students. The CoP's purpose was to identify areas within the first-year medical curriculum where topics of climate change could be integrated and to then develop a curricular thread in the first-year medical program with clear objectives and measurable outcomes. The resulting curricular thread was named Climate Change, Health, and Equity.

The goal of the Climate Change, Health, and Equity thread was to develop medical students' critical consciousness–an awareness of the impact climate change has on health that overcomes preexisting biases and is liberated from others' beliefs [19]. In addition to foundational lectures, we intended to foster the growth of medical students' critical consciousness using facilitated discussions, self-reflection and writing exercises, and experiential learning activities focused on informed advocacy, personal accountability, and environmental health justice.

## Methods

### Needs assessment and creation of the community of practice

In fall 2022, faculty members and students of the DUSOM met to identify opportunities to integrate climate change and its health impacts within the UME. The meeting resulted in the creation of an interdisciplinary CoP of medical administrators, educators, students; faculty from the Nicholas School of the Environment, Schools of Nursing and Medicine, Center for Bioethics, Humanities, and History of Medicine, and Medical Center Library; community educators from the Museum of Life and Science, the Nasher Museum of Art, and Moms Clean Air Force. Importantly, the CoP's medical educators included course directors of the first-year curriculum and medical students of all years, including student-leaders of the DUSOM Sustainability Committee and Environmental Justice and Health Equity Student Group. The CoP was tasked to develop and pilot a curricular thread, named Climate Change, Healthy and Equity, in the first-year Doctor of Medicine (MD) program by spring 2023.

Kern's six step approach to curriculum development was used to establish a needs assessment and a framework [20]. Key papers in the medical education literature were appraised for current practices, gaps, and future directions [21–23]. Educational theories of transformative learning by Mezirow and Freire [19, 24, 25], and CoP by Lave and Wenger were used to design, execute, and assess the thread [26]. Learning objectives were mapped to cognitive, affective, and psychomotor learning domains.

The curricular thread lectures were designed to provide an overview of climate change and health for all medical students. Students in the CoP also voiced the need for self-reflection activities and experiential learning opportunities that would highlight the pressing nature of climate change on the surrounding community's health and serve as tangible reasons to continue self-directed learning and informed advocacy beyond the curricular thread.

The main goals of Climate Change, Health, and Equity thread became to have medical students (1) understand the basic concepts of the planetary health framework [27]; (2) identify

effects of climate change on physical and mental health; (3) describe climate change as a social driver of health in the local community; (4) recognize examples of how climate policy disproportionately affects the health and equity of communities of color and of lower socioeconomic status; (5) recognize the importance of and learn from the lived experiences of communities disproportionately affected by climate change; (6) identify opportunities to advocate and partner with patients, community members, and policymakers to promote health equity and climate resilience; and (7) establish personal and professional accountability for continued self-directed learning.

## Thread development

After assessing the current first-year curriculum, the Climate Change, Health, and Equity thread was embedded into the required 21-week Foundations of Patient Care 2 (FPC2) course in the spring 2023 semester. Study authors AM and JV are co-directors of FPC2 and rearranged and revised content to accommodate the addition of this thread into the course.

All thread content was developed by the CoP. The thread consisted of seven lectures, including a session on critical analysis of primary literature; facilitated discussion following each lecture; two experiential learning activities in the community; and optional extracurricular opportunities (Table 1). Faculty and community experts were invited to create and deliver lectures. Each lecture contained three parts: 1) foundational knowledge, 2) the connection between climate change and health, and 3) opportunities for personal and professional advocacy and community partnership.

The lecture on critical analysis of primary literature was led by study author SK, a Research and Education Liaison Librarian at DUSOM. The objective of the session was to teach students how to effectively search interdisciplinary and subject-specific databases to identify literature on climate change and its health impacts and locate clinical decision-making tools. Following completion of the session, students were tasked with completing a self-directed learning activity in which they independently accessed, selected, and critically analyzed a primary literature source examining a lived experience related to climate. This activity aligned with LCME standard 6.3 for Self-Directed and Lifelong Learning [28], and the findings will be reported in a separate paper.

For the two experiential learning activities in the community, members of the CoP organized guided tours and presentations at the Museum of Life and Science and Nasher Museum of Art. Students were guided through special exhibits on the climate crisis. The goal of the tour was to stimulate perspective-taking by considering the diverse lived experiences within local, national, and global communities and imagining the impact of climate change on the health of subjects within art pieces and installations. Opportunities for extracurricular involvement were advertised in-class and by email, and included symposiums hosted by national and local organizations and volunteer events with the Environmental Justice and Health Equity Student Group.

## Assessment

At the conclusion of the thread, students were given two prompts designed by the CoP and intended to probe changes in preexisting beliefs and attitudes, motivation to continue learning about climate change as a social driver of health, and likelihood of participating in advocacy and community engagement. All students were required to submit a 500-word critical reflection essay. Study author SK discussed the Reflection, Evaluation for Learners' Enhanced Competencies Tool (REFLECT) rubric with students prior to the essay assignment and shared examples modeling reflective writing [29].

**Table 1. Overview of curricular thread activities.**

| Title | Lecturer/Facilitator | Format | Duration | Objectives and Major Themes | Domains |
|---|---|---|---|---|---|
| Lecture 1: Planetary Health and Disaster Justice | Brian McAdoo, Duke University Nicholas School of the Environment | Mandatory, in-person | 50 minutes | • Define disaster and planetary health<br>• Discuss direct links between planetary health and human health and wellbeing<br>• Review examples of solutions and mitigation efforts to international disasters | Cognitive, Psychomotor |
| Lecture 2: Health and Climate Change | AnnMarie Walton, Duke University School of Nursing | Mandatory, in-person | 50 minutes | • Identify health effects associated with climate change<br>• Share interdisciplinary climate education opportunities<br>• Recognize university-based efforts to address climate change<br>• Recognize contributions of healthcare to climate change<br>• Assess healthcare sustainability efforts | Cognitive, Psychomotor |
| Lecture 3: Mental Health and Climate Change | Elizabeth Bechard, Senior Policy Analyst for Moms Clean Air Force | Mandatory, in-person | | • Explore direct and indict ways climate change impacts mental health<br>• Review emerging vocabulary to describe climate change's impact on mental health<br>• vReview vulnerability factors and health disparities<br>• Review resources and potential approaches for supporting mental health and wellbeing of individuals, clinicians, and committees | Cognitive, Affective, Psychomotor |
| Lecture 4: Climate Change, Policy, and Public Health | Ashley Ward, Duke University Nicholas Institute for Energy, Environment, and Sustainability | Mandatory, in-person | 50 minutes | • Recognize heat-related illnesses in North Carolina<br>• Discuss impact of heat on pregnancy<br>• Review policy actions for extreme heat mitigation | Cognitive, Psychomotor |
| Lecture 5: Climate Change: An Ecological and Health Equity Crisis | Gaurab Basu, Cambridge Health Alliance Center for Health Equity Education and Advocacy | Mandatory, in person | 50 minutes | • Recognize the impact of climate change on health and health equity, specifically communities of color and lower socioeconomic status<br>• Identify roles clinicians take in relation to climate change<br>• Discuss sources of courage to address climate change | Cognitive, Affective, Psychomotor |
| Lecture 6: Environmental Justice and Advocacy by Students | Cameron Oglesby, Graduate Student at the Duke University Sanford School of Public Policy | Mandatory, in-person | 50 minutes | • Consider student perspectives on environmental justice during different stages of education and training<br>• Discuss tips to identifying impact | Affective, Psychomotor |
| Lecture 7: Beyond PubMed: Databases, Sources and Appraisal | Samantha Kaplan, Research and Education Liaison Librarian, Duke University School of Medicine | Mandatory, in-person | 50 minutes | • Identify library resources to support self directed learning activities<br>• Explore interdisciplinary and subject-specific databases relevant to climate change and health | Cognitive |
| Experiential Learning: North Carolina Museum of Life and Science | Max Cawley, Director of Climate Research and Engagement, North Carolina Museum of Life and Science | Mandatory, in-person | Led by museum tour guides and self-paced | • Interact with the exhibit "Imagine Durham 2100"<br>• Reflect on artwork and writing created by Durham community members and centered on hopes and anticipated needs resulting from climate change | Cognitive, Affective |
| Experiential Learning: Nasher Museum of Art | Docent of the Nasher Museum of Art | Optional, in-person, two visits offered | Self-paced | • Interact with the Spirit in the Land exhibition<br>• Consider the diverse relationships individuals have with the natural environment | Affective |

*(Continued)*

**Table 1.** (Continued)

| Title | Lecturer/Facilitator | Format | Duration | Objectives and Major Themes | Domains |
|---|---|---|---|---|---|
| Extracurricular: Embracing Health and Equity: Virtual Symposium | Carolina Advocates for Climate Change, Health, and Equity, including local healthcare providers (DO, MD, MPH, PT, PhD, RN) and trainees (residents and medical students) | Optional, virtual | 12–5 PM | • Examine cost-saving sustainable healthcare practices and carbon footprint reduction in North Carolina health systems<br>• Build a community of invested health care professionals in North Carolina<br>• Engage Health System Leaders, faculty and students, and community allies in dialogue to examine models that advance equity and sustainability | Cognitive, Affective, Psychomotor |
| Extracurricular: Tree Planting around Durham, NC | Environmental Justice and Health Equity | Optional, in-person, multiple sessions throughout the semester | | • Engage with the local community, the natural environment, and peers<br>• Consider the impact of the built environment and other social drivers of health | Psychomotor |

Prompt 1 was "Many individuals already *recognize* the importance of advocating for environmental justice and can *intellectualize* the collective responsibility needed to create change. However, "*advocating*" can be a challenging and nebulous endeavor. Please share how and if this course motivates you to act in your clinical profession and in your daily civic life." Prompt 2 was "There are communities who are disproportionately affected or will be disproportionately affected by climate change. To gain meaningful context, it is essential to learn about the lived experiences of these community members and partner with community advocates. Please share how you plan to continue engaging and learning with these individuals."

We used a sample of 30 essays, approximately 25% of the total submissions, with the expectation that this sample size would be sufficient to reach saturation, or identification of most unique themes [30–32]. Submitted essays were deidentified and assigned a number. A random number generator identified 30 numbers, and the corresponding essays were used in the analysis. Study authors TD and EA graded the selected essays according to the REFLECT rubric. As a standard setting practice, TD and EA reviewed four essays blindly and then together. The remaining essays were reviewed and graded independently; scores for each criterion were then averaged. Similarly, study authors SM and PM, who are trained in narrative medicine, performed the thematic analysis of the selected essays. Study author SM is a faculty member at DUSOM but not directly involved in this thread, and PM is a faculty member at an external United States health sciences university. SM and PM reviewed five essays together to develop a codebook. The remaining essays were reviewed and coded independently. Themes were organized according to Bloom's Taxonomy into cognitive, affective, and psychomotor domains and further classified according to the socio-ecological model for health promotion into individual, institutional, societal/community, and policy factors [33, 34].

## Study approval

The Duke University Institutional Review Board determined data collected for the Climate Change, Health, and Equity curricular thread was exempted educational research. All students participating in the curricular thread were informed of the educational research. Students were required to complete the critical reflection essay for the FPC2 course. However, they could opt out of having their essay included in this analysis by contacting an assigned individual at DUSOM, who was not involved with the thread; if the students did not contact the assigned individual, consent to have their essay analyzed was assumed. We adhered to the Standards for Reporting Qualitative Research [35].

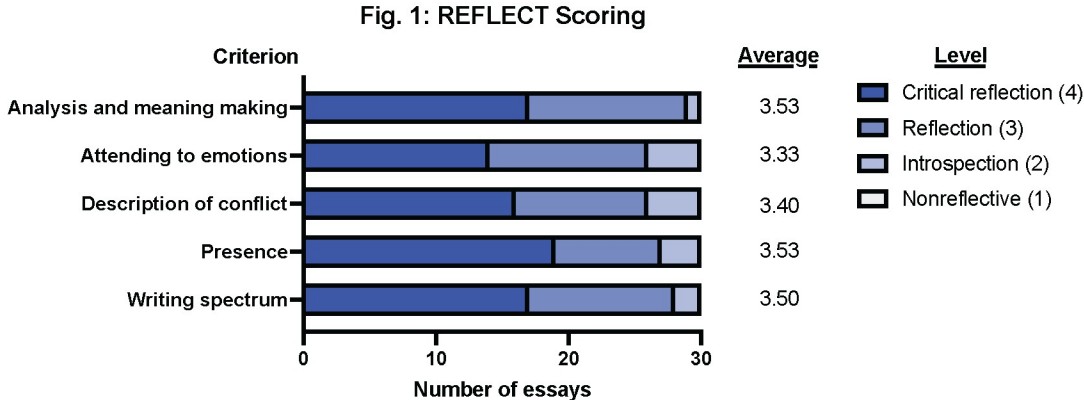

**Fig 1. REFLECT scoring of essays following participation in the Climate Change, Health, and Equity thread.** A sample of thirty essays was reviewed for elements of 1) Analysis and meaning making, 2) Attending to emotions, 3) Description of conflict, 4) Presence, 5) Writing spectrum. Criterion from each essay was scored as nonreflective (1 point), introspection (2 points), reflection (3 points), or critical reflection (4 points). Aggregate scores from all essays were averaged. For each criterion, it was found that students were either reflecting or critically reflecting. No essay was found to have non-reflective elements.

## Results

One hundred percent of the 118 first-year medical students who participated in the thread submitted an essay, and none opted out of inclusion in the analysis. Thirty of the 118 submitted essays (25% of the total sample) were randomly chosen for assessment by the REFLECT rubric and for the thematic analysis by Bloom's Taxonomy. Average scores from the REFLECT rubric highlighted that most students were reflecting or critically reflecting on thread material. None of the essays showed "nonreflective" elements (Fig 1). Major themes and supporting quotes from these essays are listed in Tables 2 to 4.

### Cognitive domain

Every student (30/30; 100%) addressed the cognitive domain (Table 2). At the individual level, students commented on newly gained vocabulary, knowledge about pathology related to heat, air quality, and infectious diseases, and perspectives on the effects of climate change on individual-level disparities. At the institutional level, students discussed how the medical field's carbon footprint contributes to climate change and how spikes in emergency visits tend to correlate with heat exposure. Students reflected on the importance of understanding the effects of climate change on the communities they are members of. Many also described how community-level health disparities they knew about previously (e.g. redlining, mental health) were linked to climate change, effectively appraising and re-evaluating their prior assumptions. Students recognized that inaction at the policy-level would contribute to widespread problems despite changes at the individual, institutional, and societal levels. They offered solutions based on their synthesized knowledge such as posing fiscal penalties on companies creating environmental hazards and taking measures for resilience and adaptation.

### Affective domain

Nearly all students (27/30; 90%) addressed the affective domain in their essays (Table 3). At the individual level, students recalled experiencing or noticing the consequences of climate change both personally and professionally in clinical settings. Students used emotionally laden words when describing their feelings on climate change mitigation efforts, including

**Table 2. Cognitive domain: Themes identified from the critical reflection essays in the Climate Change, Health, and Equity thread.**

| Factor | Theme | Representative quote |
|--------|-------|----------------------|
| **Individual** | Learning new vocabulary | "I did learn new vocabulary and specific facts that inform my thinking about climate change advocacy." |
| | Recognizing pathology associated with climate change | "Rising average temperatures, particularly at night, have been shown to have vast effects on health, including associations with preterm birth and mental health and illness." "Individuals may be impacted in indirect ways, such as the worsening air quality and changes in the spread of infectious diseases impacting the food chain and increased susceptibility to disease." "We are well-positioned to broach conversations about how climate change and zip code can impact risk of heat stroke, pregnancy complications, average air pollution exposure and risk of cardiopulmonary disease, and more." |
| | Identifying health disparities related to climate change | "I was keenly aware of the connection between climate change and health outcomes in vulnerable populations. This contrast is mainly due to my experiences living in low-income areas my entire upbringing and having seen first-hand that poor communities are almost always the first affected and bear most of the burden." "As a healthcare provider, knowing how different neighborhoods may be more or less affected by heat events, chronic diseases, access to transportation, and more can better inform the sorts of questions asked to patients and resources offered." |
| | Recognizing the daily impacts of climate change | "Before the climate change thread, I also largely thought of climate change in terms of natural disasters or dramatic weather events; however, it is critical to consider the daily impact of climate change. Thus, the climate change thread helped me expand my view to consider the daily impact of climate change on rural populations that we will be serving during our clerkship year." |
| **Institutional** | Acknowledging contribution of surgery on climate change | "...surgeons must keep climate change in mind, not because of its potential impact on patients' health, but because of the negative impact that surgery has on the environment." |
| | Correlating climate impacts with emergency medical visits | "Heat-related emergency department visits tend to spike mid-work week when people who work outside and then come home to non-air-conditioned homes begin experiencing the cumulative impacts of sustained heat exposure and lack of access to cooling at night." "...one of the most interesting things we learned this week was how many more emergency room visits there are during warm nights. It makes complete sense, but I had never thought about it before..." |
| | Acknowledging reports from expert agencies | "One major (and alarming) realization is the classification of climate change as a public health crisis by the World Health Organization (WHO)... [the] WHO reports that climate change is having a multitude of impacts on health, resulting in increased mortality and illness due to more frequent and severe extreme weather events like heatwaves, storms, and floods, as well as disturbances to food systems, a rise in zoonotic and water, food, and vector-borne diseases, and mental health problems." |
| **Societal/ Community** | Learning community-specific consequences of climate change | "I especially appreciated learning about how climate change is impacting Durham in our lecture at the museum, since this is where we'll all be living for at least the next 3.5 years... we should fully invest in the community's struggles and being part of its solutions." |
| | Redlining | "Among the new facts I learned were: redlined communities are, on average, greater than 2 degrees Celsius hotter than non-redlined communities." |
| | Realizing the widespread effects of climate change | "I hadn't realized that the direct impacts of climate change on mental health were so widespread, with heat and air pollution being associated with anxiety and other mental health disorders and 42% of Americans personally experiencing the effects of global warming." |
| | Identifying effects of climate change on equity | "This thread has been incredibly insightful as to how climate change impacts our collective health while also disproportionately disadvantaging those from lower-income areas and historically socio-politically underrepresented communities." "I appreciated learning about the concept of participatory justice, and how environmental racism refers to the exclusionary systemic practices that underrepresent and actively restrict participation from low-income, folks of color when making important decisions about waste, pollution, and other environmental issues." "I assumed that the same concept of heat as an environmental risk was most damaging to more urban populations. However, from this lecture, I learned that rural communities in North Carolina are more at risk and suffer harsher consequences." |
| | Discussing population health | "Prior to this thread, I had an understanding that climate change had drastic effects on population health and wellbeing. However, I don't think I fully understood how it is a social determinant of health." |
| **Policy** | Proposing financial penalties to benefit environmental justice | "...implementing fiscal penalties for companies that produce environmental hazards would allow funding for local communities to create their own solutions..." |
| | Highlighting infrastructure, development, and geopolitics | "Landslides have been proven to occur at rate more than two times the expected frequency within 100 meters of a road due to overly steepened slopes, poor water drainage, and inadequate debris management." |
| | Recognizing resilience and adaptation policy | "... my personal thinking about climate change has centered on crisis mitigation (ie. reducing greenhouse gases, planting more trees, switching to clean energy production, etc.) that I have largely overlooked the equally important resilience and adaptation measures that constitute essential arms of the climate crisis fight." |

**Table 3. Affective domain: Themes identified from the critical reflection essays in the Climate Change, Health, and Equity thread.**

| Factor | Theme | Representative quote |
|---|---|---|
| **Individual** | Recalling personal experiences | "I remember when I was little, [my father] went to the doctor because he was feeling unwell, and it was heating exhaustion."<br>"In 2015, Hurricane Joaquin flooded most of the state, causing many fatalities. . . My family had the resources to recover from these events, but we know many who did not. Through learning about climate change's impact on my own life, I can increase my motivation and ability to see the same effects in others." |
| | Witnessing climate-related health effects | ". . .I was waiting for beautiful lungs, but the black speckles reminded me of the damage that most humans (even healthy) will endure because of climate change and environmental hazards. The black speckles were anthracotic pigment. . . it is still quite shocking to realize that just living and existing in today's world can cause physiological changes to one's body." |
| | Feeling inadequate or fearful about climate change | "Given how dauting the challenge of fixing climate change and environmental racism is, I was initially dismayed and didn't know where to start. The magnitude of the climate change crisis and the severity of the consequences on these communities can seem insurmountable for one person.<br>"I often feel paralyzed by fear and inadequacy when I think about the magnitude of the problem and my relative powerlessness as one individual." |
| **Institutional** | Feeling frustration at scale of healthcare's carbon footprint | "I still failed to recognize the role of health systems specifically in contributing to climate change. For example, the fact that the US healthcare system alone would rank 13th in greenhouse emissions relative to all other countries is dumbfounding." |
| | Feeling frustration at lack of institutional outcomes | "It's a shame that this week could've been used to take action, such as volunteering with environmental groups, helping clean up polluted areas, working with communities to provide heat-related relief." |
| | Feeling reassured by progress in education and institutional recognition of climate change | ". . .my biggest takeaway from this thread is a sense of reassurance that climate change is being integrated into our medical education and that others in the medical community feel the same sense of urgency about it that I do." |
| **Societal/ Community** | Recognizing the impact climate change has on societal inequities | "Indigenous voices and oral traditions are unfairly left out of decisions regarding this biodiverse and culturally significant land."<br>". . .it is only on observance of vulnerable, marginalized communities, that the urgency of climate change is conveyed and with it, a humbling awareness of social and racial disparities. The visit to the Nasher Museum's exhibit, Spirit in the Land, perked my interest into the way that land holds different meaning for different communities and, secondly, how the voices of the meaning are often silenced." |
| | Feeling solidarity in community and collective action | "Seeing how many of my professors, peers, and community members care about the climate crisis has been very comforting and helped chip away at that sense of isolation. I feel the same sense of urgency as before, but a stronger sense of solidarity and empowerment."<br>"However, the thread (and my classmates) as well as outside resources, highlighted the importance that small steps and actions can have when viewed as collective actions that can be undertaken by the public." |
| **Policy** | Expressing concerns over bystander effect | "We have reached a point where most people agree on the problem, but nobody wants to take action, a global bystander effect if you will." |
| | Expressing concerns over the health effects of climate anxiety and neoliberalism | ". . .we describe Climate Anxiety faced by so many young people is the hallmark consequence of this neoliberal world view–simply that regular citizens have a duty to make choices to fix the massive problem of Climate Change. The neoliberal worldview is known to lead to worsening health outcomes for people including depression, anxiety, and perceived stress." |

"dismay", "insurmountable", "paralyzed by fear", and "powerlessness," and directed negative emotions towards the healthcare system. Students voiced critiques of this curricular thread and areas for improvement but also expressed a sense of solidarity after participating in the curricular thread alongside peers and the CoP. From the museum exhibits, students reflected on the affective importance of including perspectives that are often excluded from conversations centered around the environment, including Indigenous peoples, urban residents, and rural farmworkers. Finally, concern and anxiety regarding the lack of policy addressing climate change and its downstream effects on human health were frequently discussed.

**Table 4. Psychomotor (action) domain: Themes identified from the critical reflection essays in the Climate Change, Health, and Equity thread.**

| Factor | Theme | Representative quote |
|---|---|---|
| **Individual** | Continuing self-directed learning | "I want to keep learning about how climate change impacts individual and community health, and this thread motivated me to keep seeking out resources on my own even beyond medical school." |
| | Taking personal action | "...ways that I plan to take action on climate change will include: staying updated on local environmental concerns facing Duke's patient population, which includes my new subscription to the Durham Environmental Coalition's newsletter; continuing to vote for climate-conscious political candidates on the local, state, and national stages..." |
| | Sharing information on climate change with patients | "These are quick and easy ways that offer further information on climate change and, in turn, will allow me and those I interact with to be more adequately prepared to treat and educate patients. One way I plan to incorporate immediately is by adding ClimateRx to my badge. Having seen the effect of VoteRx on my badge–I could see ClimateRx making a similar if not larger impact on me and those who ask to learn more about the topic." |
| | Using patient demographics and tools to provide climate-informed clinical care | "I will make sure to consider each patient's social history (housing status, occupational status, etc.) as well as their potential risk factors (age, pregnancy, cardiovascular disease, etc.) to identify patients who may need additional resources on how best to protect themselves against the effects of heat-related illnesses. For patients struggling to seek shelter, I can help identify areas throughout the city that can provide shelter from extreme heat occurrences." |
| | | "I have found a database that maps out various demographic, environmental, and health statistics in the city...As a healthcare provider, knowing how different neighborhoods may be more or less affected by heat events, chronic diseases, access to transportation, and more can better inform the sorts of questions asked to patients." |
| | Environmental volunteering | "This thread motivated me to become more involved in environmental volunteering–specifically, trail days and other outdoor volunteering opportunities with the North Carolina Climbers' Coalition." |
| **Institutional** | Minimizing healthcare waste | "...healthcare systems are making an effort to implement sustainable practices like installing LEDs and reusing isolation gowns that reduce energy consumption and waste generation." <br> "The administration of the hospital and medical center should look for solutions that it can implement... Perhaps mitigating the extensive usage of air conditioning or allow the lights that are always to be cycled off unless motion is detected are the changes that could provide this support." |
| **Societal/ Community-** | Recognizing grassroots organizations | "Grassroots activism drives progressive policy changes, and progressive policy changes are often fueled by hundreds of short, seemingly unimportant meetings with legislative aide." |
| **Policy** | Advocating on policy | "...as a future physician, I see it as my role to join teams of physicians whose collective aim is to make thoughtful changes to current medical practices that will reduce carbon emissions and unnecessary energy consumption." <br> "While individual choice is important (and should be highlighted) it would be remiss to not address the fact that we can only do so much by ourselves. Our real power lies in the collective influence we can have on advocating for policies and pressuring politicians and local state officers." <br> "ways that I plan to take action on climate change will include . . . continuing to advocate for a non-profit single-payer national health program that could facilitate better care for individuals who are disproportionately affected by climate change's health impacts." |

## Psychomotor (action) domain

Nearly every essay (24/30; 80%) included statements expressing the need for action (Table 4) at the individual level (e.g. using reusable grocery bags, volunteering for creek clean-ups, talking to patients about climate change during interviews) and/or the institutional level (e.g. using light-emitting diode (LED) rather than incandescent lights, promoting sustainable transport services). Several students indicated a desire to continue learning about climate change and planetary health, through local grassroots organizations and patient-centric tools such as the Durham Neighborhood Compass, a local database that maps demographic, environmental, and health statistics according to zip code. The students conveyed that utilizing these tools could allow them to ask patients relevant questions and offer appropriate resources. Lastly, students acknowledged the power of collective influence and the sense of responsibility they felt following participation in the curricular thread to advocate for climate change policy and to vote for policies that mitigate harm to human health.

## Discussion

The Climate Change, Health, and Equity curricular thread united educators, students, and relevant community members to address the challenges of time and space in the UME in delivering content on climate change and its health impacts. Through dynamic conversations centered on the current and future impacts of climate change on health, we created foundational lectures, guided discussions, organized experiences in the community, and facilitated self-reflection on learner's attitudes and beliefs. This multi-pronged approach ultimately led students to express both the need for action and their desire to further learn from the lived experiences of patients affected by climate change in their self-reflection essays. Uniquely, we included self-reflection and experiential learning activities after students in the CoP voiced that seeing tangible effects of climate change on the health of communities may invoke desires to continue learning and advocating beyond the curricular thread.

We drew on Mezirow's transformative learning theory by establishing a learning environment that exposed students to dilemmas posed by climate change, examined preexisting belief systems, cultivated learning and perspectives, shared sentiments through discussion and critical reflection, and explored skills needed for successful reintegration in a transformed reality [24]. We applied Freire's educational learning theory to develop students' critical consciousness through creation of a communal learning environment where the teacher and learner become equals who learn from one another and the world around them [19, 25]. Utilizing a learning approach that incorporated engagement with the community brought forth the first-hand perspectives of individuals facing social, political, and economic realities of climate change. In doing so, we adhered to an essential component of Freire's theory of connecting the learners to the "others" about whom they are learning. By taking students outside of the classroom, we challenged students to disentangle biases derived from traditional narratives learned secondhand and to reframe their views based on primary accounts.

Assessments were an essential component of our thread and served two purposes. First, our evaluation tools addressed the general challenge in medical education that new innovations often lack systematic assessment. Second, our evaluation helped us determine if students were exploring their critical consciousness. The reflection essays revealed that content delivery was effective as students reported an increased cognitive understanding of the climate's impact on health. This was further contextualized by affective growth wherein students were able to connect new knowledge with existing frameworks and emotional responses. Although developing critical consciousness is a continuous process, the nearly unanimous reflections in the affective and psychomotor domains focused on responsibility and action highlight the impact our thread had on promoting introspection.

For future directions, we plan to measure the long-term impact of the thread on knowledge and action. Longitudinal follow-up is especially critical as our school's curriculum consists of only a single pre-clerkship year. Our cohort was limited to these first-year students, and we did not have a control group of students to compare reflection outcomes to. After completing future clerkships, the first-year students who participated in this thread may rethink what they have learned about climate change and health and reconsider the practicality of the behaviors and actions described in their reflections. For example, several students wanted to incorporate climate counseling into patient encounters. Follow-up after completion of the clerkships will elucidate whether counseling was feasible, and if not, what were the encountered challenges that could be addressed.

Experts have highlighted that sustainability and climate awareness are core values akin to professionalism and ethics that should be incorporated throughout training using a variety of teaching modalities as a theme rather than as standalone topics [10, 13]. To this end, we are

considering how to expand the Climate Change, Health, and Equity thread to weave throughout years of the UME. We plan to track how climate impacts on health are taught intentionally or encountered as part of the hidden UME curriculum.

Several students wanted even more diverse perspectives and a greater understanding of the impact that institutional, state, and federal policies have had on environmental justice locally. We will incorporate this feedback in future thread iterations, especially as planetary health and environmental justice encompasses efforts of all allied health professionals. Accordingly, we recommend that institutions with multiple professional schools should work towards an interprofessional framework.

Although our study was conducted at a single institution during a single year, we developed our thread such that it could be readily adapted at other health professional programs. We recognize that the impacts of climate change and resulting inequities have regional differences, such as distribution of infectious disease, and community-specific variation based on pre-existing conditions of the local population, the nearby environment, and risk exposure [13]. Other health professional programs can adapt the didactic materials and the experiential community-based activities to provide meaningful learning opportunities that can be tailored locally [36, 37]. For example, the Museum of Life and Science and Nasher Museum of Art tours that were organized for this thread could be exchanged for immersive, creative experiences at neighborhood community centers disproportionately affected by climate change, local farms and food distribution and recovery programs, waste and treatment facilities, and fieldwork sites to appreciate how environmental health is surveilled. These experiences would provide health professional students opportunities to learn about local planetary health priorities and to reflect critically on practices back in traditional training spaces.

We view the education provided by the Climate Change, Health, and Equity thread as imperative to medical students' education, personal growth, and professional responsibility to their patients. We will continue developing our first-year thread by incorporating insights gained from this cohort, extending educational activities into subsequent years of UME, and partnering with allied health professional programs and institutions.

## Author Contributions

**Conceptualization:** Trisha Dalapati, Sneha Mantri, Phillip Mitchell, Kathryn M. Andolsek, J. Matthew Velkey, Jennifer Lawson.

**Data curation:** Trisha Dalapati, Emily J. Alway.

**Investigation:** Trisha Dalapati, Sneha Mantri, Ian A. George, Jennifer Lawson, Andrew J. Muzyk.

**Methodology:** Sneha Mantri, Samantha Kaplan, Andrew J. Muzyk.

**Project administration:** Samantha Kaplan.

**Supervision:** Andrew J. Muzyk.

**Visualization:** Emily J. Alway.

**Writing – original draft:** Trisha Dalapati, Ian A. George, Samantha Kaplan, Andrew J. Muzyk.

**Writing – review & editing:** Trisha Dalapati, Emily J. Alway, Ian A. George, Andrew J. Muzyk.

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
