## [Decision Letter · Decision Letter 0]

27 Dec 2023

PONE-D-23-31947Fostering medical students’ critical consciousness on climate change, health, and equityPLOS ONE

Dear Dr. Muzyk,

Thank you for submitting your manuscript to PLOS ONE. You will read that two of the reviewers were quite critical about your manuscript. If you feel that you can answer the questions that they have raised, then we invite you to submit a revised version of the manuscript.

We look forward to receiving your revised manuscript.

Kind regards,

Frederick Grinnell

Academic Editor

PLOS ONE

5. Please ensure that you include a title page within your main document. You should list all authors and all affiliations as per our author instructions and clearly indicate the corresponding author.

6. Please amend your authorship list in your manuscript file to include authors Andrew Muzyk, Trisha Dalapati, Sneha Mantri, Phillip Mitchell, Ian Allen George, Kathryn Andolsek, John Matthew Velkey, and Jennifer Lawson.

7. Please amend either the abstract on the online submission form (via Edit Submission) or the abstract in the manuscript so that they are identical.

8. Please include your full ethics statement in the ‘Methods’ section of your manuscript file. In your statement, please include the full name of the IRB or ethics committee who approved or waived your study, as well as whether or not you obtained informed written or verbal consent. If consent was waived for your study, please include this information in your statement as well.

Reviewers' comments:

Reviewer's Responses to Questions

**Comments to the Author**

1. Is the manuscript technically sound, and do the data support the conclusions?

Reviewer #1: Partly

Reviewer #2: Yes

Reviewer #3: Partly

2. Has the statistical analysis been performed appropriately and rigorously? 

Reviewer #1: I Don't Know

Reviewer #2: N/A

Reviewer #3: No

3. Have the authors made all data underlying the findings in their manuscript fully available?

Reviewer #1: Yes

Reviewer #2: Yes

Reviewer #3: No

4. Is the manuscript presented in an intelligible fashion and written in standard English?

Reviewer #1: Yes

Reviewer #2: Yes

Reviewer #3: Yes

5. Review Comments to the Author

Reviewer #1: Overall, this paper has utility for educators at other institutions interested in creating more climate content for their students. It is unclear how much of this one example is applicable generally, since a unique feature of the effort at this institution is that the existing course directors played a key part in introducing new content.

More importantly, the information about the course may be too scant to be useful. And not enough detail warrants much more detail, which isn’t feasible in this sort of paper. I would thus streamline the paper to the crucial, transferable content, eliminating details that are not relevant to the overall architecture of the course, which will also make this paper more readable. I hope that my specific comments can provide some additional guidance.

Of note, the writing could use the librarian’s once-over. “At present” not “presently”. “Effect” not “affect”; “synthesize: not “synthesize ..together” etc. And that’s just the opening section

p. 1 “surrounding environment” This is vague and can be anything. Please be clear about what aspects of the surrounding environment you mean, as this orients your argument.

p. 2

Building teaching capacity… financial support” Consider including:

Blanchard OA, Greenwald LM, Sheffield PE. The Climate Change Conversation: Understanding Nationwide Medical Education Efforts. Yale J Biol Med. 2023 ;96(2):171-184.

“Though leaders have recommended learning objectives…” Consider including: ]

Sorensen C. Campbell H, Depoux A. et al. Core Competencies to Prepare Health Professionals to Respond to the Climate Crisis. Plos Climate 2023; 2(6): e0000230.

“Considering the education need and expert recommendation” It’s very important to be precise here, as this represents the transition from the previous general overview/state of climate education to this institution’s response. What exactly was the CoP convened to address? Start this paragraph with “In order to address…” Then state what the goal of the CoP was (e.g. In order to address the need for a more robust climate curriculum with measurable goals, we…”

How were the learning objectives and goals derived (based on what sources?)

Materials and institutional expertise: vague

Assessment tools: Unclear how these were developed, given that no intervention has been developed, according to this description

Where does the Health and Equity come from? This hasn’t been part of the intro. Is this what was meant as part of the ‘education need and expert recommendation?” Also, it is hard to understand how the group, given the authors’ recognition of all the constraints preventing climate education, managed to get this thread into the curriculum. What seems to be needed here is more focus. Instead of starting the paper with all the problems, why not say something like (in a nutshell): “Although many institutions have confronted difficulties in incorporating climate change awareness into their busy curriculum, our medical school’s first-year climate, health, and equity course successfully addressed this barrier by integrating multiple urgent concerns students are confronting today. These include environmental threats, health equity, and [whatever other urgency the thread contains].”

p. 3 evolves from? You must mean ‘evolves beyond’ or something like that

“We intended…humanities” The reader recognizes this as a big ask and hopes that what follows will illuminate how these goals were addressed

p. 4. The power of the course directors is key here; the lack thereof would be a major obstacle at other institutions

“118 students engaged in thread content”: isn’t there a better way to say this? How about: “The course thread included…” you can mention the 118 students later, probably not necessary until the results

“All thread content was developed” It seems as though you are going backwards at best or all over the place at worst. Reorder to state: Who developed thread content, how it was integrated (as lectures etc); and additional content (2 mandatory sessions; 2 community engagement sessions).

Interactive and community sessions: It is unclear how these matter to the focus of your paper. You say too little about their relevance. I would just mention them, maybe with the briefest possible mention in the context of the overall climate and equity intention of the course. All you need to say without getting into deep water is “Exhibits addressing the [physical and emotional?] impacts of climate change on affected communities were well-suited to provoking thoughtful discussions.”

Overall: consider what the reader needs to understand about the basic course structure and intent. In terms of all the details, the reader won’t have the same local resources you have; if they think they might be able to apply your approach to their own resources, they can contact you.

It’s unclear how beliefs about climate change are a ‘metareflective exercise of their learning.” The reader has no idea from this sentence what your goal is. Also if they were required to write this, how come they had the option to opt out? It sounds contradictory. Maybe: any student who did not choose to opt out was required…

p. 7 There seems to be an overlap between methods and results. Methods should state what the overall plan and process is. The results should state how many people participated, if any did not participate, and what was found.

Nothing is said about the thematic analysis itself and how it was conducted. See: Kiger ME, Varpio L. Thematic analysis of qualitative data: AMEE Guide No. 131. Med Teach. 2020;42(8):846-854.

7-8 “For instance, one student connected a lecture statistic regarding the nature of poor air quality ..with environmental pathology…” The reader is unlikely to understand what the actual insight here is, so I suggest finding a better, less obscure-sounding example

“…expand their medical knowledge…to include a critical review of the literature” It’s unclear how the above examples, which are pretty basic, show this. Wouldn’t students have heard in their sessions about the disparities mentioned in the quoted material?

9 ”…importance of hearing directly from affected communities” It’s unclear whether students were asking for such contacts, or whether in the course they had such contacts. Or were they responding to the museum exhibit?

Psychomotor: It’s unclear how much content in the teaching thread addressed actionable items

10 “Students participated in foundational lectures…on attitudes and beliefs” I feel that this should be embedded in the method section or tabulated in the results section. This is not the place to mention this as though it were some sort of concluding insight

“resulted in calls to action” There is a big difference between calls to action as expressed in an assigned essay vs in an activist format. “Students reflected in their essays on the different ways students could actively engage” seems more accurate. Better yet: “These activities ultimately led students to express a need for action and a desire to learn from the lived experience of patients affected by climate change. Through critical reflection…”

However, I don’t think you can claim that students generated a sense of personal & professional responsibility…etc. The more accurate claim would be that they expressed this sense.

It’s unclear to me how the CoP transformed the topic of climate change. Nothing you describe really supports this. And for whom do you mean? For the first year students or the CoP members? If the latter, you don’t show it in the paper. If the former, all we have is the claims in their essays. Maybe the students did express something like “This course really transformed my thinking” but you don’t provide evidence of that. If you mean the CoP members, we really don’t hear anything in this paper about the impact on them.

“Holistic educational approach towards addressing inequities…” This doesn’t seem to have been the goal of this paper as stated in the intro

“Didactics were augmented…etc” Why is this in the discussion? This seems like it should be in methods or not directly relevant to the paper.

p. 11 What are ‘unanimous reflections?” Do you mean, nearly all participants highlighted the impact the thread had in furthering reflection on the roles of privilege, responsibility and action?

Future Directions: Condense the first three sentences to ‘long-term impact’ on knowledge and action, as these plans seem unformed/speculative as yet

Adapting experiential community-based activities: I don’t know if the readers really understand what the authors did in their museum visits. The alternative suggestions are potentially great, but it’s not that easy for educators to adapt them, especially without knowing the details of what the specifics of such community exhibits are—not to mention in what context they could make demands of this sort on the curriculum. Maybe a Table detailing examples and specific, itemized suggestions would help

Reviewer #2: I commend the authors for their great work on addressing such a timely and relevant topic. Please see some suggestions below:

Minor grammatical suggestions: Add a comma in the first line of the abstract.

…impact on the environment and health, medical schools...

Page 1: Education on climate change’s affect (change to effect) on health...

Methods: How and why are 30 essays chosen for thematic analysis? Is 30 a representative number of essays here? Would suggest to add more information regarding this in the methods section.

Reviewer #3: 1.

General question -- The invitation from PLOS ONE reads as follows:

I am inviting you to peer review the following manuscript for PLOS ONE. In view of your expertise, I would greatly value your input on this manuscript’s suitability for publication.

Manuscript #: PONE-D-23-31947

Title: Fostering medical students’ critical consciousness on climate change, health, and equity

Authors: Andrew Muzyk; Trisha Dalapati; Sneha Mantri; Phillip Mitchell; Ian Allen George; Kathryn Andolsek; John Matthew Velkey; Jennifer Lawson

Article type: Research Article

So, why are you anonymizing the manuscript, e.g., Study authors XX and XX were responsible for the creation and implementation of the Climate Change, Health, and Equity thread in the first-year medical school pre-clerkship curriculum at the XXSOM and elsewhere. Makes the manuscript harder to read.

2. Table 1 provides a summary but no details regarding the cognitive, affective, and psychomotor domains and further classified according to the socio-ecological model for health promotion into individual, institutional/relationship, societal, and policy factors.

A. The manuscript itself should include quantitative data regarding frequencies.

B. Either the manuscript or an appendix should include actual examples that were interpreted according to the different categories.

3. What is the control? Are the students’ answers a consequence of the Climate Change, Health, and Equity thread in the first-year medical school pre-clerkship curriculum or would 2nd year or more advanced students who didn’t participate in the thread have provided similar answers?

6. PLOS authors have the option to publish the peer review history of their article (what does this mean?). If published, this will include your full peer review and any attached files.

Reviewer #1: No

Reviewer #2: No

Reviewer #3: No

---

## [Author Response · Author response to Decision Letter 0]

16 Apr 2024

The Response to Reviewers and Letter have been uploaded as part of the files. I am also including the responses here. I apologize for any formatting errors as these comments were tabulated in the file. 

--------

Dear Dr. Chenette, 

We are pleased to submit our revised manuscript now entitled, “Development of a curricular thread to foster medical students’ critical reflection and promote action on climate change, health, and equity.” 

We are greatly appreciative of the comments we received from the three reviewers. As we hope you will see, by incorporating suggestions, we have significantly improved our manuscript. 

Our manuscript focuses on a timely curricular innovation undertaken at our medical school – the incorporation of climate change, health, and equity topics into undergraduate medical education. We describe the needs assessment; development of a community of practice; and design and execution of curricular materials, including lectures, discussions, experiential learning opportunities in the community, and a critical reflection essay. A total of 118 first-year medical students participated in the thread and we thoroughly analyzed the critical reflection essays for themes and by the REFLECT rubric. 

While revising this manuscript, we have now included substantial details in our methods and results sections, included suggested citations by the reviewers, and revised the flow of the writing. Most notably, we have 1) added one figure summarizing the results of the critical reflection essays according to the REFLECT rubric and 2) included four tables. The first table outlines all our curricular activities in detail, as requested by the reviewers, including the lecture title, lecturer name and affiliation, learning objectives, and Bloom’s Taxonomy domains addressed. The remaining three tables conveys the themes found in the critical reflection essays according to Bloom’s Taxonomy and the socio-ecological model. We have included numerous direct quotes to convey the themes and highlight the efficacy that students felt this thread had on their learning, reflection, and growth. We hope that by including all new materials, readers may consider implementing this thread at their own institutions, which is a necessity given the pressing nature of climate change and its health impacts.

Our findings highlight that medical students synthesize information about climate change into their medical knowledge and can reflect on the affective and actionable elements of climate change in relation to human health. We wish to underscore not only the enthusiasm of students who participated but also the student-driven nature of the thread development itself and writing of this manuscript. Altogether, our curricular thread depicts the fruitful roles the next generation of physicians are taking in further medical education innovations. 

The total word count of our abstract is 296 words. Our manuscript is approximately 3,400 words (excluding abstract, references, tables, figure, and appendix). All authors on this manuscript have contributed substantially to the work including conceptualizing the purpose, reviewing the literature, and writing the manuscript. None have conflicts of interest to declare.

Sincerely, 

Trisha Dalapati 

Medical Student 

Duke University School of Medicine

--------

We acknowledge that we have reviewed the PLOS ONE style requirements. 

We thank the editor for this comment and agree wholeheartedly with the importance of making raw data accessible. It was not within our IRB to publish the full essays of the student-participants in this thread. Therefore, to be as transparent as possible, we have added three tables (Table 2-4) that shows numerous direct quotes from student essays, which support our findings and conclusions. 

We appreciate this comment by the reviewers. We have added more detail regarding the participant consent in the Methods section. It reads as follows: 

“The Duke University Institutional Review Board determined data collected for the Climate Change, Health, and Equity curricular thread was exempted educational research. All students participating in the curricular thread were informed of the educational research. Students were required to complete the critical reflection essay for the FPC2 course. However, they could opt out of having their essay included in this analysis by contacting an assigned individual at DUSOM, who was not involved with the thread; if the students did not contact the assigned individual, consent to have their essay analyzed was assumed. We adhered to the Standards for Reporting Qualitative Research.”

We request to revise this statement to “All relevant data are within the manuscript.” We have included our qualitative data in Tables 2-4. It was not within our IRB to publish the full essays of the student-participants in this thread. These three tables (Table 2-4) that shows numerous direct quotes from student essays, which support our findings and conclusions.

5. Please ensure that you include a title page within your main document. You should list all authors and all affiliations as per our author instructions and clearly indicate the corresponding author.

We acknowledge that we have done this. 

6. Please amend your authorship list in your manuscript file to include authors Andrew Muzyk, Trisha Dalapati, Sneha Mantri, Phillip Mitchell, Ian Allen George, Kathryn Andolsek, John Matthew Velkey, and Jennifer Lawson.

We acknowledge that we revised our authorship list and included the title page within the manuscript file. 

7. Please amend either the abstract on the online submission form (via Edit Submission) or the abstract in the manuscript so that they are identical.

We acknowledge that we revised our abstract and included the revised version within the manuscript file.

8. Please include your full ethics statement in the ‘Methods’ section of your manuscript file. In your statement, please include the full name of the IRB or ethics committee who approved or waived your study, as well as whether or not you obtained informed written or verbal consent. If consent was waived for your study, please include this information in your statement as well.

We have included the following in our manuscript Methods section: 

“The Duke University Institutional Review Board determined data collected for the Climate Change, Health, and Equity curricular thread was exempted educational research. All students participating in the curricular thread were informed of the educational research. Students were required to complete the critical reflection essay for the FPC2 course. However, they could opt out of having their essay included in this analysis by contacting an assigned individual at DUSOM, who was not involved with the thread; if the students did not contact the assigned individual, consent to have their essay analyzed was assumed. We adhered to the Standards for Reporting Qualitative Research.”

1. Is the manuscript technically sound, and do the data support the conclusions?

We appreciate the reviewers’ comments. We acknowledge that in several areas of our results we could have included more direct quotes to support our findings and conclusions. We also wholeheartedly agree with the reviewer’s comments that we should have included more details on the curricular activities, including lectures, to facilitate reproducing this type of curricular thread at another institution. To address these important concerns, we have made a table to outline the thread activities (Table 1) and tables of direct quotes from student essays to underscore the thematic analysis conclusions. 

2. Has the statistical analysis been performed appropriately and rigorously? 

We appreciate the reviewer’s comments. We agree with reviewer 2 that this comment is not largely applicable to our research study which is qualitative in nature. 

3. Have the authors made all data underlying the findings in their manuscript fully available?

We recognize that Reviewer 3 did not feel that we adequately made all underlying findings available. As previously mentioned, we have now included our qualitative data in Tables 2-4. It was not within our IRB to publish the full essays of the student-participants in this thread. We hope that addition of these three tables (Table 2-4) showing direct quotes from student essays will address reviewer 3’s concerns. 

Reviewer #1: 

Overall, this paper has utility for educators at other institutions interested in creating more climate content for their students. It is unclear how much of this one example is applicable generally, since a unique feature of the effort at this institution is that the existing course directors played a key part in introducing new content.

More importantly, the information about the course may be too scant to be useful. And not enough detail warrants much more detail, which isn’t feasible in this sort of paper. I would thus streamline the paper to the crucial, transferable content, eliminating details that are not relevant to the overall architecture of the course, which will also make this paper more readable. I hope that my specific comments can provide some additional guidance.

We thank the reviewer for recognizing the utility of this manuscript in helping other institutions create climate content for their students. Certainly, we acknowledge the key role of the course directors in developing and presenting this content to students. We hope that this manuscript can serve as a guide to creating a similar climate thread at other institutions, lowering the barrier to institute new content, and making it easier for course directors and other institutional leaders to present similar thread content to students.

For other institutions, we agree that detail is critical for reproducibility and even improvement upon the thread that we have created. For this reason, we have edited the manuscript throughout to emphasize the transferable content. Additionally, we have added more detail to Table 1 (“Overview of curricular thread activities”), which now includes lecturer affiliations, session lengths, as well as objectives, themes, and Bloom’s Taxonomy domains addressed by each thread component. We have addressed specific comments below.

Reviewer #1: 

Of note, the writing could use the librarian’s once-over. “At present” not “presently”. “Effect” not “affect”; “synthesize: not “synthesize ..together” etc. And that’s just the opening section

We thank the reviewer for pointing out these errors. The manuscript has been thoroughly revised for grammar and clarity. 

Reviewer #1: 

p. 1 “surrounding environment” This is vague and can be anything. Please be clear about what aspects of the surrounding environment you mean, as this orients your argument.

Thank you for this comment. We have specified that “surrounding environment, including the climate, geography, and healthcare inequities in the community.” 

Reviewer #1: 

Building teaching capacity… financial support” Consider including:

Blanchard OA, Greenwald LM, Sheffield PE. The Climate Change Conversation: Understanding Nationwide Medical Education Efforts. Yale J Biol Med. 2023 ;96(2):171-184.

We thank the reviewer for pointing us to this important paper. We have added in Blanchard et al.’s article as reference 12. 

Reviewer #1: 

“Though leaders have recommended learning objectives…” Consider including: ]

Sorensen C. Campbell H, Depoux A. et al. Core Competencies to Prepare Health Professionals to Respond to the Climate Crisis. Plos Climate 2023; 2(6): e0000230.

We thank the reviewer for pointing us to this important paper. We have added in Sorensen et al.’s article as reference 18.

Reviewer #1: 

“Considering the education need and expert recommendation” It’s very important to be precise here, as this represents the transition from the previous general overview/state of climate education to this institution’s response. What exactly was the CoP convened to address? Start this paragraph with “In order to address…” Then state what the goal of the CoP was (e.g. In order to address the need for a more robust climate curriculum with measurable goals, we…”

We strongly agree with the reviewer’s comment that we were initially not clear nor precise with the goal of the CoP. Therefore, we have revised the paragraph to address this issue:

“In fall 2022, faculty members and trainees met to address the lack of UME at our institution, Duke University School of Medicine (DUSOM), focused on climate change and its impacts on health. Subsequently, we created a community of practice (CoP) of medical educators and students. The CoP’s purpose was to identify areas within the first-year medical curriculum where topics of climate change could be integrated and to then develop a curricular thread in the first-year medical program with clear objectives and measurable outcomes. The resulting curricular thread was named Climate Change, Health, and Equity.”

Reviewer #1: 

How were the learning objectives and goals derived (based on what sources?)

In Table 1 (“Overview of Curricular Thread Activities”), we outline the curricular objectives and themes from each activity that was planned for the curricular thread. These learning objectives and goals were based on each lecturer’s expertise and were reviewed by the larger CoP. 

Reviewer #1: Materials and institutional expertise: vague

We greatly thank the reviewer for this comment. We have included information specifying the affiliation of each member of the CoP in the text of the manuscript and included Table 1 to specify the materials, expertise, objectives, and goals used for the curricular thread. 

“In fall 2022, faculty members and students of the DUSOM met to identify opportunities to integrate climate change and its health impacts within the UME. The meeting resulted in the creation of an interdisciplinary CoP of medical administrators, educators, students; faculty from the Nicholas School of the Environment, Schools of Nursing and Medicine, Center for Bioethics, Humanities, and History of Medicine, and Medical Center Library; community educators from the Museum of Life and Science, the Nasher Muse

---

## [Editor Report · Decision Letter 1]

29 Apr 2024

Development of a curricular thread to foster medical students’ critical reflection and promote action on climate change, health, and equity

PONE-D-23-31947R1

Dear Dr. Muzyk,

We’re pleased to inform you that your manuscript has been judged scientifically suitable for publication and will be formally accepted for publication once it meets all outstanding technical requirements.

Kind regards,

Frederick Grinnell

Academic Editor

PLOS ONE
---

## [Editor Report · Acceptance letter]

17 May 2024

PONE-D-23-31947R1 

PLOS ONE

Dear Dr. Muzyk, 

I'm pleased to inform you that your manuscript has been deemed suitable for publication in PLOS ONE. Congratulations! Your manuscript is now being handed over to our production team.

Kind regards, 

on behalf of

Dr. Frederick Grinnell 

Academic Editor

PLOS ONE